# 360-InpaintR: Reference-Guided 3D Inpainting for Unbounded Scenes

## Abstract

This paper introduces 360-InpaintR, the first reference-based 360° inpainting method for 3D Gaussian Splatting (3DGS) scenes, particularly designed for unbounded environments. Our method leverages multi-view information and introduces an improved unseen mask generation technique to address the challenges of view consistency and geometric plausibility in 360° scenes. We effectively integrate reference-guided 3D inpainting with diffusion priors to ensure consistent results across diverse viewpoints. To facilitate research in this area, we present a new 360° inpainting dataset and capture protocol, enabling high-quality novel view synthesis and quantitative evaluations of modified scenes. Experimental results demonstrate that 360-InpaintR performs favorably against existing methods in both quantitative metrics and qualitative assessments, particularly in complex scenes with large view variations.

## 1 Introduction

Three-dimensional scene reconstruction and manipulation, revolutionized by Neural Radiance Fields (NeRFs) and their extensions, are crucial for various applications like VR/AR, robotics, and autonomous driving. A key challenge is removing objects from 3D scenes while realistically filling the resulting holes, which is valuable for real estate visualization, augmented reality, and computer vision preprocessing. However, reference-based inpainting in 3D Gaussian Splatting (3DGS) scenes, especially in 360° unbounded environments, remains challenging. This task requires exploiting multi-view information, filling never-observed areas, and maintaining consistency and geometric plausibility across views.

Figure 1 illustrates our pipeline for reference-based 360° unbounded scene inpainting. Given input images with camera parameters, object masks, and a reference image, we generate a 3D Gaussian Splatting (3DGS) representation for novel view rendering. Our method exploits multi-view information and leverages generative processes to fill unseen areas, ensuring inpainted regions are coherent, plausible, and consistent across views. By combining 3DGS's multi-view consistency with 2D in-

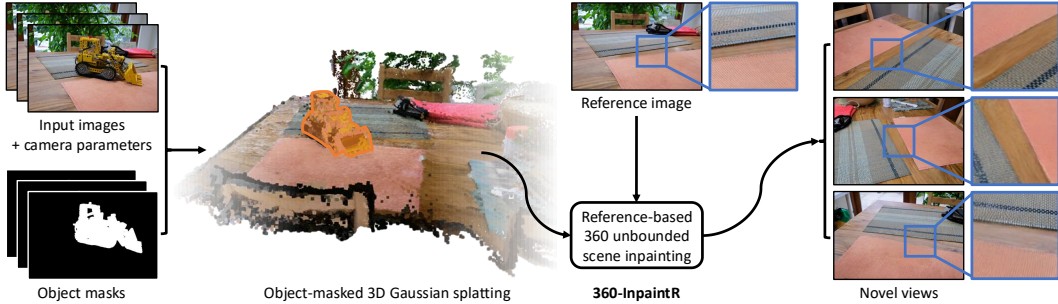

Figure 1: **Overview of our reference-based 360° unbounded scene inpainting method.** Given input images with camera parameters, object masks, and a reference image, our 360-InpaintR approach generates an object-masked 3D Gaussian Splatting representation. This representation can then render novel views of the inpainted scene, effectively removing the masked objects while maintaining consistency with the reference image.

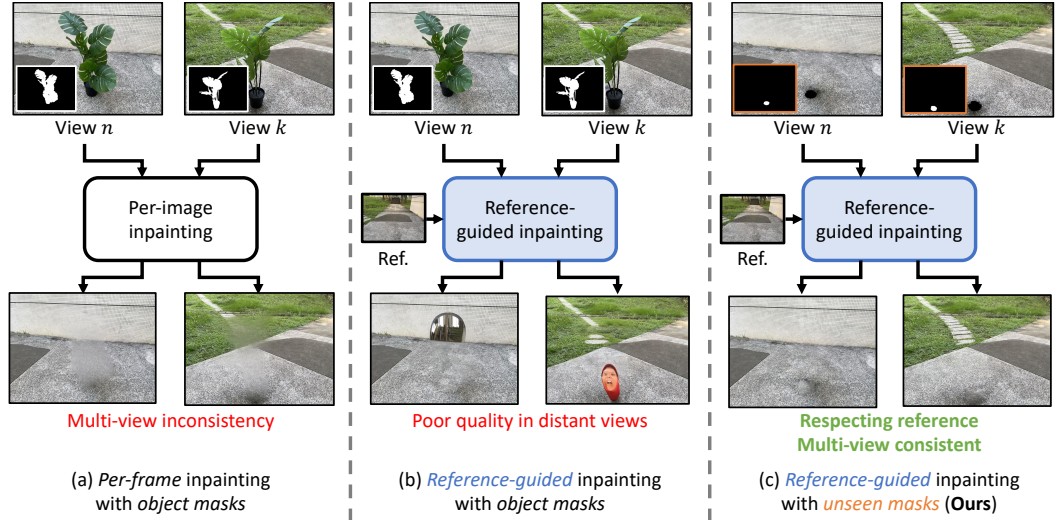

Figure 2: **Comparison of different inpainting approaches for 3D scenes.** (a) Per-frame inpainting with object masks leads to multi-view inconsistencies. (b) Reference-guided inpainting with object masks improves consistency but results in poor quality for views distant from the reference. (c) Our approach using reference-guided inpainting with unseen masks respects the reference view while maintaining multi-view consistency, addressing the limitations of previous methods.

painting models' generative power, we address challenges in view consistency and 3D geometry, especially for significant view changes.

Figure 2 illustrates key challenges in 3D scene inpainting. Per-frame approaches (a) lead to multi-view inconsistencies, while reference-guided methods (b) struggle with distant views due to hallucinations from inpainting models like Stable Diffusion. Our approach (c) uses unseen masks to maintain consistency across views while respecting the reference. Existing methods face significant limitations. InNeRF360 (Wang et al., 2023b) underutilizes multi-view information, missing valuable contextual cues. Gaussian Grouping (Ye et al., 2024), while effective at object removal, struggles with 3D consistency and risks over-inpainting due to tracking errors. SPIn-NeRF (Mirzaei et al., 2023b) and LaMa-based (Suvorov et al., 2022) methods face challenges with view consistency, especially in complex scenes or large view variations. These shortcomings underscore the need for a more robust approach to 3D scene inpainting that maintains consistency, preserves geometric accuracy, and adapts to the challenges of 360° unbounded environments.

Our goal is to develop a comprehensive 3D scene inpainting method that respects the reference view, maintains 3D consistency, and leverages multi-view background information. Given posed RGB images and a reference image, we generate an inpainted 3D Gaussian Splatting (3DGS) representation with view consistency. Figure 2 illustrates our approach's advantages. We address limitations of per-frame inpainting (a) and reference-guided inpainting (b) by using unseen masks (c), effectively leveraging multi-view information. Our method handles 360° unbounded environments with dramatic view changes and high scene complexity. By integrating advanced inpainting with 3DGS, we produce geometrically accurate, visually plausible results that blend seamlessly with the original scene, enabling high-quality novel view synthesis even in challenging scenarios.

The key contributions of our work include:

- The first reference-based 360° inpainting method for 3DGS scenes, leveraging multi-view information with improved unseen mask generation.

- An effective integration of reference-guided 3D inpainting and diffusion priors for consistent results across diverse viewpoints.

- A comprehensive framework including a new 360° inpainting dataset and capture protocol, enabling high-quality novel view synthesis and quantitative evaluations of modified scenes.

## 2 RELATED WORK

### 2.1 RADIANCE FIELDS FOR NOVEL VIEW SYNTHESIS

**NeRF.** Neural Radiance Fields (NeRF) (Mildenhall et al., 2020) have revolutionized novel view synthesis, combining differentiable volume rendering (Tulsiani et al., 2017; Henzler et al., 2019) and positional encoding (Vaswani et al., 2017; Gehring et al., 2017) to implicitly represent 3D scenes. Subsequent works have improved efficiency (Liu et al., 2020; Garbin et al., 2021; Yu et al., 2021a), quality (Barron et al., 2021b; Zhang et al., 2020), and data requirements (Yu et al., 2021b; Wang et al., 2021). While NeRF excels in view synthesis, editing and manipulating NeRF scenes, especially for tasks like object removal and inpainting, remains challenging. Recent works have explored object editing (Yang et al., 2021; Yuan et al., 2022), stylization (Wang et al., 2023a), and limited inpainting (Liu et al., 2022; Mirzaei et al., 2023b), but consistent, high-quality 3D inpainting in complex NeRF scenes remains an open problem.

**3D Gaussian splatting.** 3D Gaussian Splatting (Kerbl et al., 2023) offers an efficient alternative to NeRF (Mildenhall et al., 2020), representing scenes as explicit 3D Gaussians. This approach enables faster rendering and training (Mildenhall et al., 2020), handles multi-scale representations (Barron et al., 2021a), and facilitates easier scene editing (Liu et al., 2021). Recent extensions include dynamic scene modeling (Yang et al., 2024b), semantic incorporation (Chen et al., 2024), and combinations with diffusion models (Wynn & Turmukhambetov, 2023), advancing novel view synthesis and scene manipulation.

### 2.2 2D IMAGE INPAINTING

**Traditional methods.** Image inpainting has evolved from early PDE-based techniques (Bertalmio, 2000) to exemplar-based methods (Criminisi et al., 2004). Texture synthesis (Efros & Leung, 1999) and patch-based approaches like PatchMatch (Barnes et al., 2009) further advanced the field. Despite limitations with large missing regions and complex textures (Jam et al., 2021; Liu et al., 2018), these methods established principles now incorporated into learning-based approaches (Liu et al., 2018; Yu et al., 2019). Their computational efficiency remains valuable in resource-constrained scenarios (Jam et al., 2021).

**Deep learning-based methods.** Deep learning has revolutionized image inpainting, with CNNs like Context Encoders (Pathak et al., 2016) pioneering the field. GANs (Goodfellow et al., 2014) and models like DeepFillv2 (Yu et al., 2019) further improved results. Large Mask Inpainting (LaMa) (Suvorov et al., 2022) addressed large missing regions effectively. Recently, diffusion models (Ho et al., 2020), particularly Stable Diffusion (Rombach et al., 2022), have shown remarkable capabilities, leveraging complex data distributions (Dhariwal & Nichol, 2021). While these methods have significantly improved inpainting quality, challenges remain (Li et al., 2023). This success has inspired 3D inpainting research (Liu et al., 2022; Prabhu et al., 2023), though extending 2D approaches to 3D presents unique challenges (Mirzaei et al., 2023a).

**Reference-based methods.** Reference-based inpainting methods (Zhao et al., 2022) address limitations of general inpainting approaches by utilizing additional visual information. LeftRefill (Tang et al., 2023) exemplifies this approach, using a two-stage architecture with feature matching and refinement networks. These methods offer greater user control and diverse outputs (Zhao et al., 2022), showing promise in various applications (Jam et al., 2021). However, challenges remain in seamless integration and reference selection (Li et al., 2023). The success of these methods has inspired extensions to 3D inpainting tasks (Liu et al., 2022; Prabhu et al., 2023), although adapting to 3D presents unique challenges (Mirzaei et al., 2023a).

### 2.3 3D SCENE INPAINTING

**Methods without multi-view background knowledge.** Early 3D inpainting approaches extended 2D concepts to 3D without extensive multi-view knowledge. These include direct 3D shape completion methods like PCN (Yuan et al., 2018), 2.5D representations (Shih et al., 2020), and generative models like 3D-GAN (Wu et al., 2016). In neural rendering, EditNeRF (Liu et al., 2021) and NeRF-In (Liu et al., 2022) pioneered NeRF editing and inpainting. These methods often struggle with view consistency (Mirzaei et al., 2023b) and global context (Wang et al., 2023b). Despite limitations, they laid groundwork for more advanced, multi-view aware techniques (Mirzaei et al., 2023a).

Figure 3: **Overview of our method.** Our approach takes multi-view RGB images and corresponding object masks as input and outputs a 3D Gaussian Splatting (3DGS) representation with the masked objects removed. The pipeline consists of three main stages: (a) Unseen Masks Generation using depth warping to detect truly occluded areas, (b) Depth-Aware 3DGS Initialization to fill disocclusion regions after object removal, and (c) Reference-Guided Inpainting and 3DGS Finetuning, which iteratively refine the 3DGS representation using a reference-based 2D diffusion inpainting model and ensure multi-view consistency.

**Methods leveraging multi-view information.** Multi-view 3D inpainting methods address limitations of single-view approaches. SPIn-NeRF (Mirzaei et al., 2023b) combines NeRF with multi-view image inpainting. Philip & Drettakis (2018) use multi-view stereo for object removal in image-based rendering. Inpaint3D (Prabhu et al., 2023) leverages learned 3D priors. InpaintNeRF360 (Wang et al., 2023b) extends to 360-degree scenes, while Gaussian Grouping (Ye et al., 2024) uses 3D Gaussian Splatting. These methods maintain consistency across viewpoints (Mirzaei et al., 2023a) but face challenges with large-scale occlusions (Weder et al., 2023), computational costs (Barron et al., 2023), and view inconsistencies (Yin et al., 2023). Despite challenges, they advance scene editing and completion, potentially leading to new applications (Bommasani et al., 2021).

## 3 METHOD

Our method takes multi-view RGB images $\{I_n\}$ and object masks $\{M_n\}$ as input, where $n \in [1..N]$. It outputs a 3D Gaussian Splatting (3DGS) representation with masked objects removed. As shown in Figure 3, our approach has three stages: (1) Unseen Masks Generation using depth warping, (2) Depth-Aware 3DGS Initialization leveraging monocular and incomplete depth, and (3) Reference-Guided Inpainting and 3DGS Finetuning using a 2D diffusion model. This process effectively propagates textures across views in unbounded scenes, resulting in high-quality, consistent 3D inpainting.

### 3.1 UNSEEN MASKS GENERATION

Accurately identifying regions requiring inpainting is crucial for maintaining scene consistency and maximizing the use of available background information. Our unseen mask generation approach addresses two main scenarios: identifying areas without Gaussians after removal and detecting regions where inappropriate Gaussians become visible.

**Identifying regions using the seen attribute.** We introduce a seen attribute $v_i$ for each Gaussian $i$ in the scene. During training, we optimize this attribute using the following loss:

$$\mathcal{L}_{\text{seen}} = \sum_n \sum_p |R_v(p, n) - 1|, \tag{1}$$

where $R_v(p, n)$ is the rendered seen attribute at pixel $p$ in view $n$, and the target value is 1 for all pixels. After removing Gaussians with the mask attribute, we generate an initial unseen mask $U_{\text{init}}$

Figure 4: **Unseen mask generation process using depth warping.** The rendered depth $D_n$ and object mask $M_n$ from view $n$ are warped to view $i$ using camera poses. The warped mask $M_{n \to i}$ is compared with the object mask $M_i$ in view $i$. Through backward traversal and aggregation across multiple views, we obtain the unseen mask $U$ for view $n$. The refined unseen mask $U_{\text{refine}}$ is generated by applying average and threshold operations to the aggregated mask.

for each view $n$:

$$U_{\text{init}}(p, n) = \begin{cases} 1 & \text{if } R_v(p, n) < \tau_{\text{init}}, \\ 0 & \text{otherwise,} \end{cases} \tag{2}$$

where $\tau_{\text{init}}$ is a threshold value, typically set to a small positive number (*e.g.*, 0.1).

**Depth warping for detecting inappropriate Gaussians.** To refine the unseen mask, we employ a depth warping technique. Figure 4 illustrates the process of generating the unseen mask using depth warping. For each view $n$, we compute:

$$U_{\text{refine}}(p, n) = \begin{cases} 1 & \text{if } \left( \frac{1}{K-1} \sum_{i \neq n} M_i(\mathcal{W}(p, D_n, T_{n \to i})) \cap M_n(p) \right) > \tau_{\text{refine}}, \\ 0 & \text{otherwise,} \end{cases} \tag{3}$$

where $K$ is the number of views, $M_i$ is the object mask for view $i$, $D_n$ is the depth map for view $n$ after object removal, $T_{n \to i}$ is the transformation from view $n$ to view $i$, and $\mathcal{W}(p, D, T)$ is a warping function that projects pixel $p$ using depth $D$ and transformation $T$, and $\tau_{\text{refine}}$ is a threshold value.

**Combining the approaches.** Our final unseen mask effectively captures both areas without Gaussians and regions with inappropriate Gaussians:

$$U_{\text{final}}(p, n) = \max(U_{\text{init}}(p, n), U_{\text{refined}}(p, n)). \tag{4}$$

This mask $U_{\text{final}}$ is then used in subsequent stages of our pipeline to guide the inpainting process, ensuring that we focus on areas truly requiring reconstruction while preserving as much original scene information as possible. We provide complete steps of the unseen masks generation algorithm in the supplementary materials.

## 3.2 DEPTH-AWARE 3DGS INITIALIZATION

After completing object removal and unseen mask generation, we proceed to initialize the 3D Gaussian Splatting (3DGS) in the disocclusion regions. This process is crucial for ensuring a coherent and realistic reconstruction of the inpainted areas.

**Using monocular depth and rendered incomplete depth.** We begin by selecting a reference view. For this view, we can render incomplete RGB image $I_{\text{ref}}^{\text{inc}}$ and incomplete depth map $D_{\text{ref}}^{\text{inc}}$. Our initialization process consists of the following steps. First, We apply an RGB inpainting method to $I_{\text{ref}}^{\text{inc}}$ to obtain a complete RGB image $I_{\text{ref}}^{\text{comp}}$. Next, using the inpainted RGB image, we estimate a monocular depth map using Depth Anything V2 (Yang et al., 2024a): $D_{\text{ref}}^{\text{mono}} =$

DepthAnythingV2($I_{\text{ref}}^{\text{comp}}$). Then, to ensure consistency between the estimated monocular depth and the incomplete rendered depth, we perform depth alignment using Poisson image editing (Pérez et al., 2023): $D_{\text{ref}}^{\text{aligned}} = \text{PoissonImageEdit}(D_{\text{ref}}^{\text{mono}}, D_{\text{ref}}^{\text{inc}})$. This aligned depth map combines the completeness of the monocular depth estimation with the accuracy of the rendered incomplete depth in the known regions.

**Initializing 3DGS in disocclusion regions.** With the aligned depth map $D_{\text{ref}}^{\text{aligned}}$, we proceed to initialize new Gaussians in the disocclusion regions. First, we unproject the aligned depth map to 3D space, focusing on the disocclusion regions identified by the unseen mask. This unprojection takes into account the camera's intrinsic parameters. For each pixel $(u, v)$ in the unseen region where $U_{\text{final}}(u, v) = 1$, we compute the 3D point $P = (X, Y, Z)$ as follows:

$$Z = D_{\text{ref}}^{\text{aligned}}(u, v), X = (u - c_x) \cdot Z/f_x, Y = (v - c_y) \cdot Z/f_y, \tag{5}$$

where $(f_x, f_y)$ are the focal lengths in pixels and $(c_x, c_y)$ are the principal point offsets. This process gives us a set of initial 3D points $P$. Next, We use these unprojected points $P$ as initial positions for new Gaussians in the disocclusion regions. Finally, the existing Gaussians from the background (*i.e.*, those not removed in the object removal step) are kept fixed during this initialization and the following optimization process. This initialization strategy, incorporating accurate camera intrinsics, provides a geometrically correct starting point for the subsequent fine-tuning of the 3DGS representation. It ensures that the newly added Gaussians in the disocclusion regions are consistent with both the inpainted RGB content and the surrounding geometry while respecting the proper 3D spatial relationships defined by the camera model.

### 3.3 Reference-Guided Inpainting and 3DGS Finetuning

After initializing the trainable 3D Gaussian Splatting (3DGS), we need to finetune it using inpainted RGB images. We leverage the multi-view consistency capability of reference-guided 3D inpainting models by using the selected reference view's RGB image as the input reference, which then inpaints all other training views. These inpainted views serve as our ground truth for finetuning the 3DGS. We employ LeftRefill (Cao et al., 2024), a reference-guided diffusion model, as our 2D inpainting model. LeftRefill reformulates reference-based synthesis as a contextual inpainting process. It stitches reference and target views as $I' = [I_{\text{ref}}; \hat{I}_{\text{tar}}] \in \mathbb{R}^{H \times 2W}$, where $I_{\text{ref}}$ is the reference image, $\hat{I}_{\text{tar}}$ is the masked target image, and $H$ and $W$ are the height and width of the images. LeftRefill employs task and view-specific prompt tuning optimized by: $p_t, p_v{}^* = \arg\min_{p_t, p_v} \mathbb{E}[|\varepsilon - \varepsilon_\theta([z_t; \hat{z}_0; M], c_\phi(p_t, p_v), t)|^2]$, where $p_t$ and $p_v$ are task and view prompt embeddings, $\varepsilon_\theta(\cdot)$ is the estimated noise by the Latent Diffusion Model (Rombach et al., 2022), $c_\phi(\cdot)$ is the frozen CLIP-H (Radford et al., 2021), $z_t$ is a noisy latent feature, $\hat{z}_0$ are masked latent features, and $M$ is the mask.

Once we have generated inpainted RGB images for all training views, we use these as supervision to finetune our 3DGS. During the finetuning process, we only update the Gaussians that were unprojected in the depth-aware initialization step. The other Gaussians that were retained during the object removal stage remain fixed. To finetune our 3DGS, we use a combination of L1 loss and LPIPS (Learned Perceptual Image Patch Similarity) (Zhang et al., 2018) loss. The total loss for finetuning is formulated as:

$$\mathcal{L} = \mathcal{L}_1 + \lambda_{\text{LPIPS}} \mathcal{L}_{\text{LPIPS}}. \tag{6}$$

### 3.4 Implementation Details

In implementation, we utilize an L1 loss to optimize both masked attributes and the seen attribute. The learning rate is set to 0.1 for both. When training masked attributes, we follow GaussianGrouping Ye et al. (2024), which enforces a constraint that adjacent GS-masked attributes should exhibit a smaller loss. This ensures that masked attributes are effectively removed. For the threshold value $\tau_{\text{init}}$ and $\tau_{\text{refine}}$, we set it to 0.5 and 0.35. For the inapinting stage, we employed the masked LPIPS loss derived from the SPIN-NeRF framework to mitigate the proliferation of floaters. We empirically set $\lambda_{\text{LPIPS}}$ to 0.5 and fine-tune 3DGS for 10,000 iterations in our experiments.

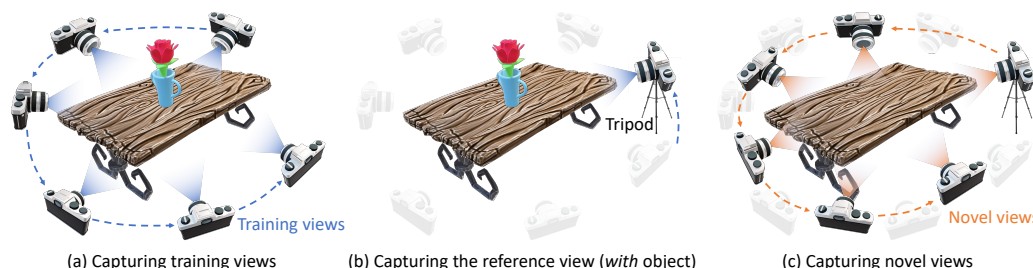

(a) Capturing training views     (b) Capturing the reference view (*with* object)     (c) Capturing novel views

Figure 5: **Illustration of the data capture process for the 360-USID dataset.** (a) Capturing training views: Multiple images are taken around the object in the scene. (b) Capturing the reference view: A camera is mounted on a tripod to capture a fixed reference view (with an object). (c) Capturing novel views: After removing the object, additional images are taken from various viewpoints, including one from the same tripod position as the reference image.

## 4 360° UNBOUNDED SCENES INPAINTING DATASET (360-USID)

To address the lack of publicly available reference-based 360° inpainting datasets for evaluation, we introduce the 360° Unbounded Scenes Inpainting Dataset (360-USID), comprising seven scenes.

### 4.1 DATASET COLLECTION PROTOCOL

We developed a protocol using a standard camera to create this dataset, as simultaneously capturing multi-view photos with and without objects is challenging and typically requires specialized equipment. Our protocol, illustrated in Figure 5, consists of:

1. Placing an object (*e.g.*, a vase) on a textured surface in a 360° unbounded scene and capturing 200-300 photos around it as input training images.

2. Mounting the camera on a tripod and capturing one final training view with a fixed position and angle.

3. Removing the object and capturing novel view photos from the same tripod position for ground truth evaluation. Other novel view positions differ from training views.

To ensure high-quality captures, we select surfaces with rich textural details, stabilize the tripod, and disable white balance. We record video and extract the sharpest frames using the variance of the Laplacian method to minimize motion blur. Each scene comprises 200-300 training images and around 30 testing images for quantitative evaluations. Consistent lighting is maintained to minimize the impact of object shadows on reference and testing images.

### 4.2 SCENE DESCRIPTIONS

Our 360-USID dataset, shown in Figure 6, features seven diverse scenes: four outdoor (Box, Cone, Lawn, Plant) and three indoor (Cookie, Sunflower, Dustpan). These scenes present various challenges for 3D inpainting tasks, representing a range of real-world environments. Each scene has 171-347 training views and 31-33 ground truth novel views. Most scenes are captured at 960×540 resolution, with Plant and Dustpan at 960×720. This diversity in content, view counts, and resolutions makes 360-USID a robust tool for evaluating 3D inpainting algorithms in complex scenarios.

### 4.3 DATA PREPROCESSING AND CAMERA POSE ESTIMATION

For data preprocessing and camera pose estimation, we employ the following steps:

1. We use COLMAP (Schönberger & Frahm, 2016; Schönberger et al., 2016) or a similar Structure-from-Motion (SfM) pipeline such as hloc (Sarlin et al., 2019; 2020) or GLOMAP (Pan et al., 2024) to compute a shared 3D coordinate space for both training and novel views.

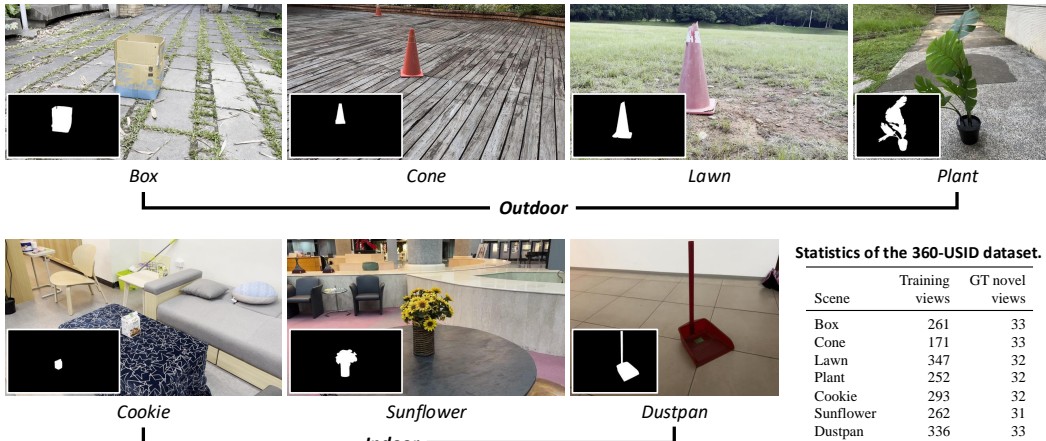

| Statistics of the 360-USID dataset. | | |
|---|---|---|
| Scene | Training views | GT novel views |
| Box | 261 | 33 |
| Cone | 171 | 33 |
| Lawn | 347 | 32 |
| Plant | 252 | 32 |
| Cookie | 293 | 32 |
| Sunflower | 262 | 31 |
| Dustpan | 336 | 33 |

Figure 6: **Overview of the 360-USID dataset.** Sample images from each scene, including four outdoor scenes (Box, Cone, Lawn, Plant) and three indoor scenes (Cookie, Sunflower, Dustpan). (*Bottom right*) The table shows statistics for each scene, including the number of training views and ground truth (GT) novel views. The dataset provides a diverse range of environments for evaluating 3D inpainting methods in both indoor and outdoor settings.

2. As the object is removed in novel views, we generate object masks using SAM 2 (Segment Anything in Images and Videos) (Ravi et al., 2024) and input these into COLMAP to ignore object reconstruction.

3. After obtaining camera poses for training and novel views from COLMAP, we can input the training images into any NeRF/3DGS inpainting method to remove the object.

4. We then use these methods' resulting radiance fields or 3D representations to render novel view photos, which we compare against our captured ground truth novel view images for quantitative evaluation.

## 5 EXPERIMENTS

### 5.1 EXPERIMENTAL SETUP

**Datasets.** We evaluate our method on two types of 360° unbounded environment datasets:

- **360-USID (Ours)**: We introduce a new dataset specifically for evaluating 360° unbounded scene inpainting. It comprises 7 scenes (3 indoor, 4 outdoor), each with 200-300 training views containing the object to be removed and about 30 test views without the object. This dataset provides ground truth for quantitative evaluation of 360° inpainting tasks. We maintain the width at 960 pixels when evaluating 360-USID to preserve high-fidelity details crucial for 360° scene representation.

- **MipNeRF-360 (Barron et al., 2022) and NeRFStudio (Tancik et al., 2023)**: We use these established 360° datasets to demonstrate our method's performance on additional unbounded scenes. We evaluate at 1/4 resolution to balance computational efficiency with performance. While lacking ground truth for inpainting evaluation, these datasets are valuable for qualitative assessments and demonstrating our method's generalization to various complex, unbounded environments.

**Metrics.** To evaluate our 360° inpainting method, we employ two primary metrics that focus on the perceptual quality and realism of the inpainted scenes:

- **LPIPS (Learned Perceptual Image Patch Similarity) (Zhang et al., 2018)**: This perceptual metric measures the similarity between the inpainted renderings and ground-truth images. Lower values indicate better perceptual similarity.

Table 1: **Quantitative comparison of 360° inpainting methods on the 360-USID dataset.**

| LPIPS ↓ / FID ↓ | Box | Cone | Lawn | Plant | Cookie | Sunflower | Dustpan | Average |
|---|---|---|---|---|---|---|---|---|
| Gaussian Grouping | 0.447 / 109.991 | 0.380 / 129.666 | 0.578 / 295.259 | 0.378 / 66.919 | 0.638 / 429.472 | 0.415 / 205.217 | 0.225 / 117.099 | 0.437 / 193.375 |
| LeftRefill | 0.508 / 97.989 | 0.369 / 114.716 | 0.621 / 188.048 | 0.677 / 160.690 | 0.676 / **232.896** | 0.501 / 108.186 | 0.344 / **100.718** | 0.528 / 143.321 |
| 3DGS + LaMa | 0.470 / 99.414 | 0.395 / **107.241** | 0.545 / 213.319 | 0.553 / 119.828 | **0.559** / 251.464 | 0.518 / **62.026** | 0.263 / 104.697 | 0.472 / 136.855 |
| 3DGS + LeftRefill | 0.535 / 102.535 | **0.317** / 117.106 | 0.603 / 210.115 | 0.593 / 264.670 | 0.577 / 461.026 | 0.407 / 98.044 | **0.210** / 140.723 | 0.463 / 199.174 |
| Ours | **0.344 / 88.937** | 0.340 / 122.200 | **0.453 / 178.767** | **0.333 / 56.710** | 0.568 / 243.332 | **0.337** / 84.864 | 0.315 / 165.075 | **0.384 / 134.270** |

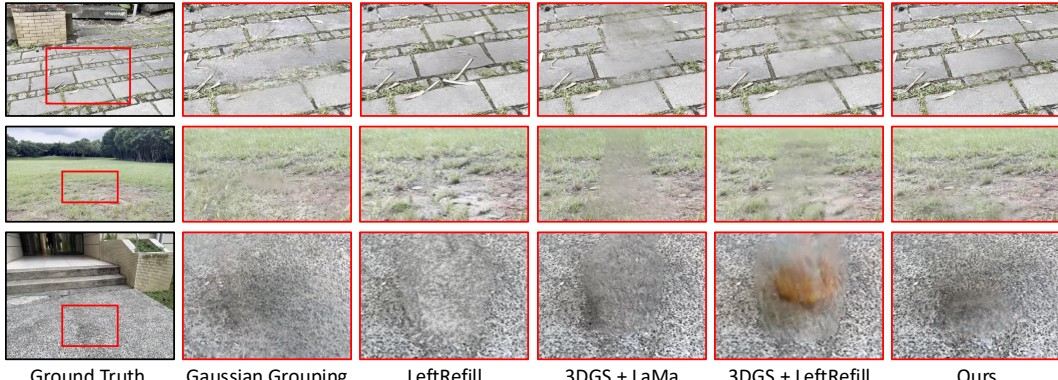

Ground Truth    Gaussian Grouping    LeftRefill    3DGS + LaMa    3DGS + LeftRefill    Ours

Figure 7: **Qualitative comparison of 360° inpainting methods on the 360-USID dataset.**

- **FID (Fréchet Inception Distance) (Heusel et al., 2017)**: This metric assesses the statistical similarity between the distribution of features from inpainted and ground-truth images. Lower FID scores indicate higher fidelity and realism of the inpainted regions.

For both LPIPS and FID, we compute the metrics only within the inpainted regions. This approach, similar to that used in SPIn-NeRF (Mirzaei et al., 2023b), allows us to focus specifically on the quality of the inpainting rather than the overall scene reconstruction. For the 360-USID dataset, where we have ground-truth images without the removed objects, we compute both LPIPS and FID. For MipNeRF-360 and NeRFStudio datasets, which lack ground truth for inpainting, we rely on qualitative assessments. We provide additional evaluation results using PSNR and SSIM (Wang et al., 2004) in the supplementary materials for a more comprehensive analysis.

## 5.2 COMPARISONS WITH STATE-OF-THE-ART METHODS

**Quantitative comparisons.** We evaluate 360-InpaintR against state-of-the-art approaches on the 360-USID dataset. Table 1 shows LPIPS and FID scores across different scenes. Our method consistently outperforms existing approaches. Gaussian Grouping (Ye et al., 2024) struggles with 360° consistency, while LeftRefill (Cao et al., 2024) improves but falls short in 360° environments. 3DGS + LaMa (Suvorov et al., 2022) and 3DGS + LeftRefill show better results than 2D methods but face view consistency challenges. 360-InpaintR achieves the lowest average LPIPS and FID scores, indicating superior perceptual quality and similarity to ground truth. The performance gap is particularly notable in scenes with complex geometry or large removed objects, highlighting our method's ability to leverage multi-view information and maintain 360° consistency.

**Qualitative visual comparisons.** Figure 7 compares our 360-InpaintR method against state-of-the-art approaches on challenging scenes from 360-USID, Mip-NeRF360, and NeRFStudio datasets. Our method excels in maintaining view consistency and preserving fine details in 360° unbounded environments. While Gaussian Grouping and LeftRefill show strengths in object removal and 2D inpainting, respectively, they struggle with 360° scene consistency. 3DGS + LaMa and 3DGS + LeftRefill improve upon 2D methods but face challenges with complex geometries and large inpainted regions. 360-InpaintR consistently produces sharper, more detailed, and view-consistent results across all scenes, effectively handling challenging cases like periodic textures and complex organic structures. It preserves fine details, overall scene structure, and view-dependent effects crucial for 360° scene realism, particularly in varying lighting conditions or reflective surfaces. We provide additional video results in our supplementary materials.

Table 2: **Ablation study of our 360-InpaintR method.**

| Unseen mask | Depth initialization | 2D inpainter | LPIPS ↓ | FID ↓ |
|:---:|:---:|:---:|:---:|:---:|
| - | ✓ | LeftRefill | 0.022 | 181.177 |
| ✓ | - | LeftRefill | 0.020 | 139.511 |
| ✓ | ✓ | LaMa | 0.020 | 179.912 |
| ✓ | ✓ | LeftRefill | **0.019** | **134.268** |

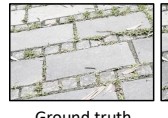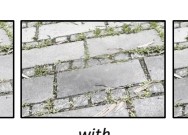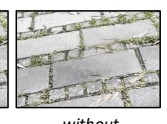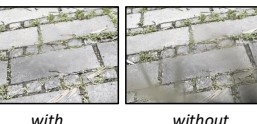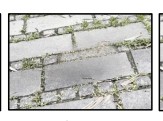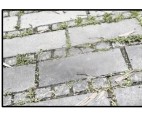

Ground truth · *with* · *without* · *with* · *without* · *with* LaMa · *with* LeftRefill

Unseen mask · Depth initialization · 2D inpainter

Figure 8: **Qualitative comparisons of ablation studies.**

## 5.3 ABLATION STUDIES

To evaluate the effectiveness of each component in our 360-InpaintR method, we conduct a series of ablation studies. Table 2 presents the quantitative results of these studies, and Figure 8 shows qualitative comparisons.

**Unseen mask generation.** We compare our unseen mask generation technique with directly using object masks. Our approach significantly improves inpainting quality, particularly in areas occluded from multiple views. The unseen masks help to identify truly occluded regions, leading to more accurate and consistent inpainting results. This is especially noticeable in scenes with complex geometries, where object masks alone may not capture all the necessary information for effective inpainting.

**Effect of depth-aware 3DGS initialization.** The depth-aware 3DGS initialization proves crucial for maintaining geometric consistency in the inpainted regions. Compared to random initialization, our method produces more structurally coherent results, especially in areas with significant depth variations. This is particularly evident in scenes where the inpainted geometry needs to blend seamlessly with the existing scene structure.

**Inpainting method comparison.** We evaluate the performance of two inpainting methods: LaMa (Suvorov et al., 2022) for per-image inpainting and LeftRefill (Cao et al., 2024) for reference-guided inpainting. While both methods show improvements over baseline approaches, LeftRefill consistently outperforms LaMa in our 360° setting. This is due to LeftRefill's ability to leverage information from the reference view, leading to more consistent results across different viewpoints. However, combining either method with our full pipeline still outperforms their standalone usage.

## 6 CONCLUSION

We presented 360-InpaintR, a novel reference-based 360° inpainting method for 3D Gaussian Splatting scenes in unbounded environments. Our approach effectively addresses the challenges of object removal and hole filling in complex 3D scenes. Key contributions include leveraging multi-view information through improved unseen mask generation, integrating reference-guided 3D inpainting with diffusion priors, and introducing the 360-USID dataset for comprehensive evaluation. Experimental results demonstrate 360-InpaintR's superior performance over existing methods, particularly in complex geometries and large view variations. While this work represents a significant advancement in 3D scene editing, future directions include improving computational efficiency, handling dynamic scenes, and integrating more advanced language models for intuitive editing.

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

# A APPENDIX

## A.1 UNSEEN MASKS GENERATION ALGORITHM

We provide detailed steps of the unseen masks generation algorithm in Algorithm A.1.

---

**Algorithm 1** Unseen Masks Generation

---

**Input:** Set of views $V = v_1, ..., v_K$, object masks $M = M_1, ..., M_K$, removal depths $D = D_1, ..., D_K$, transformations $T = T_{i \to j} | i, j \in [1, K], i \neq j$

**Output:** Final unseen masks $U_{\text{final}} = U_{\text{final}}1, ..., U\text{final}_K$

1: // Train seen attribute
2: **for** each training iteration **do**
3:      Render seen attribute $R_v(p, n)$ for all pixels $p$ and views $n$
4:      Compute $\mathcal{L}\text{seen} = \sum n \sum_p |R_v(p, n) - 1|$
5:      Update seen attribute based on $\mathcal{L}_{\text{seen}}$
6: **end for**
7: // Generate unseen masks
8: **for** $n = 1$ to $K$ **do**
9:      // Initialize mask using seen attribute
10:      **for** each pixel $p$ **do**
11:          $U_{\text{init}}(p, n) \leftarrow \begin{cases} 1 & \text{if } R_v(p, n) < \tau_{\text{init}} \\ 0 & \text{otherwise} \end{cases}$
12:      **end for**
13:      // Refine mask using depth warping
14:      $U_{\text{refined}}(p, n) \leftarrow 0$ for all pixels $p$
15:      **for** $i = 1$ to $K$, $i \neq n$ **do**
16:          $M_{n \to i} \leftarrow \mathcal{W}(M_n, D_n, T_{n \to i})$
17:          **for** each pixel $p$ **do**
18:              **if** $p \in M_{n \to i} \cap M_i$ **then**
19:                  $U_{\text{refined}}(p, n) \leftarrow U_{\text{refined}}(p, n) + 1$
20:              **end if**
21:          **end for**
22:      **end for**
23:      $U_{\text{refined}}(p, n) \leftarrow U_{\text{refined}}(p, n)/(K - 1)$ for all pixels $p$
24:      $U_{\text{refined}}(p, n) \leftarrow \begin{cases} 1 & \text{if } U_{\text{refined}}(p, n) > \tau_{\text{refine}} \\ 0 & \text{otherwise} \end{cases}$
25:      // Combine approaches
26:      **for** each pixel $p$ **do**
27:          $U_{\text{final}}(p, n) \leftarrow \max(U_{\text{init}}(p, n), U_{\text{refined}}(p, n))$
28:      **end for**
29: **end for**
30: **return** $U_{\text{final}}$

---

## A.2 ADDITIONAL QUANTITATIVE EVALUATIONS

We provide additional quantitative evaluations using PSNR and SSIM in Table 3 and Table 4.

Table 3: **PSNR comparison of 360° inpainting methods on the 360-USID dataset.**

| PSNR ↑ | Box | Cone | Lawn | Plant | Cookie | Sunflower | Dustpan | Average |
|---|---|---|---|---|---|---|---|---|
| Gaussian Grouping | 15.485 | 13.010 | 13.537 | 16.139 | 11.984 | 19.267 | 22.150 | 15.939 |
| LeftRefill | 15.867 | 13.996 | 14.667 | 12.815 | 9.102 | 14.437 | 21.644 | 14.647 |
| 3DGS + LaMa | 15.230 | 13.305 | 15.515 | 12.919 | 10.215 | 12.183 | 22.308 | 14.525 |
| 3DGS + LeftRefill | 15.013 | 14.083 | 14.712 | 13.702 | 9.990 | 18.138 | 22.411 | 15.436 |
| Ours | 15.851 | 13.922 | 16.109 | 17.358 | 10.063 | 19.304 | 22.815 | 16.489 |

Table 4: **SSIM comparison of 360° inpainting methods on the 360-USID dataset.**

| SSIM ↑ | Box | Cone | Lawn | Plant | Cookie | Sunflower | Dustpan | Average |
|---|---|---|---|---|---|---|---|---|
| Gaussian Grouping | 0.967 | 0.977 | 0.992 | 0.909 | **0.980** | **0.989** | 0.993 | **0.972** |
| LeftRefill | 0.948 | 0.961 | 0.979 | 0.822 | 0.948 | 0.967 | 0.986 | 0.944 |
| 3DGS + LaMa | 0.967 | **0.980** | 0.992 | 0.879 | 0.976 | 0.987 | **0.994** | 0.968 |
| 3DGS + LeftRefill | 0.968 | 0.979 | 0.992 | 0.873 | 0.971 | 0.982 | 0.992 | 0.965 |
| Ours | **0.971** | **0.980** | **0.993** | **0.919** | 0.976 | 0.919 | **0.994** | 0.966 |

