# OpenReview forum: "360-InpaintR: Reference-Guided 3D Inpainting for Unbounded Scenes"
_ICLR.cc/2025/Conference — ICLR 2025 Conference Withdrawn Submission_

### Official Review · Reviewer_6SGh · 2024-11-01

**Soundness:** 2
**Presentation:** 3
**Contribution:** 1
**Rating:** 1
**Confidence:** 5

**Summary:**

This paper proposes an inpainting method for scenes reconstructed using Gaussian Splatting (3DGS). They suggest a three-stage solution to solve this:
1. In the first stage, they propose the "Unseen Mask Generation" technique to address the view consistency challenge and detect areas that require inpainting.
2. In the second stage, they prepare inpainted RGB and depth image for the reference view and initialize the Gaussian primitives.
3. In the final stage, they finetune the 3DGS model using inpainted RGB images.

Further, they propose a 360-USID dataset which consists of seven scenes.

**Strengths:**

1. **Dataset Contribution**: This work proposes the first dataset to evaluate inpainting for $360^{\circ}$ unbounded scenes for NeRF and Gaussian Splatting based methods. 360-USID is an object-centric dataset and comprises seven scenes.

**Weaknesses:**

1. **Limited Technical Contribution**:
  - It is unclear how the method for generating view-consistent masks differs from SPiN-NeRF [W1] and InNeRF360 [W2]. I request that authors provide a detailed comparison between their view-consistent mask generation method and those used in SPiN-NeRF and InNeRF360. Highlight the key differences and potential advantages of the proposed approach.
  - Similarly, utilizing the depth of the unmasked region and inpainting, the incomplete depth map is also discussed in methods InNeRF360[W2] and DateNeRF[W3].
  - Also, using pre-trained diffusion models for inpainting is also discussed in DateNeRF[W3],  InFusion[W4], InNeRF360[W2] etc.

2. The motivation behind the proposed method needs further clarity. The authors have not addressed the limitations of existing methods or explained how this approach differs from these methods. These points remain unclear in the current manuscript. I recommend that the authors address these aspects in the Introduction section. I suggest that the authors provide a brief comparison table highlighting the critical limitations of existing methods and how their approach addresses each of these limitations.

3. **Claim of the first reference-based method**: This claim might not be correct. As there already exists a method by  Mirzaei et al.[W5] that uses a single reference image for inpainting.
Limited Comparison: The authors did not compare with SPIN-NeRF[W1],   InNeRF360[W2], InFusion[W4] and Mirzaei et al.[W5].

4. **Artifacts in the final output**: There are visible artifacts in the inpainted scenes : "render_traj_color_sunflower_compressed.mp4", "render_traj_color_garden_compressed.mp4", "render_traj_color_dustpan_compressed.mp4", "render_traj_color_cookie_compressed.mp4", "render_traj_color_cone_compressed.mp4" and "render_traj_color_bonsai_compressed.mp4" shared in "inpaint" folder in the supplementary material. Is this a limitation of the method? Please discuss these artifacts during the rebuttal phase.


[W1] Mirzaei, Ashkan, et al. "SPIn-NeRF: Multiview segmentation and perceptual inpainting with neural radiance fields." Proceedings of the IEEE/CVF Conference on Computer Vision and Pattern Recognition. 2023.

[W2] Wang, Dongqing, et al. "InNeRF360: Text-Guided 3D-Consistent Object Inpainting on 360-degree Neural Radiance Fields." Proceedings of the IEEE/CVF Conference on Computer Vision and Pattern Recognition. 2024.

[W3] Rojas, Sara, et al. "DATENeRF: Depth-Aware Text-based Editing of NeRFs." arXiv preprint arXiv:2404.04526 (2024).

[W4] Liu, Zhiheng, et al. "InFusion: Inpainting 3D Gaussians via Learning Depth Completion from Diffusion Prior." arXiv preprint arXiv:2404.11613 (2024).

[W5] Mirzaei, Ashkan, et al. "Reference-guided controllable inpainting of neural radiance fields." Proceedings of the IEEE/CVF International Conference on Computer Vision. 2023.

**Questions:**

1. What are the scenarios where existing methods fail? How does the proposed method solve these issues? Please explain these differences clearly.

---

### Official Review · Reviewer_1Rm2 · 2024-11-03

**Soundness:** 2
**Presentation:** 1
**Contribution:** 1
**Rating:** 3
**Confidence:** 5

**Summary:**

This paper introduces 360-InpaintR, a novel approach for reference-guided 3D inpainting in 360° unbounded scenes using 3DGS. The method removes objects from 3D scenes while filling in the resulting gaps in a geometrically consistent and view-coherent manner. Key contributions include an innovative unseen mask generation process, depth-aware initialization for realistic geometry reconstruction, and integration with a reference-guided 2D inpainting model to maintain multi-view consistency. The results show that 360-InpaintR outperforms existing methods in visual quality and geometric accuracy, especially in complex 3D scenes. Additionally, the authors introduce a new dataset, 360-USID, to support quantitative evaluation and research in this domain.

**Strengths:**

1. This paper introduces a framework for 3D inpainting tailored for unbounded, 360° environments, addressing challenges such as object removal and filling of unseen regions while maintaining geometric plausibility and multi-view consistency. This task is important for scene editing applications in AR/VR.
2. Depth-Aware Gaussian Spawning: By using depth-aware initialization for 3D Gaussians after depth inpainting with monocular depth estimation priors, the method ensures that newly generated Gaussians align geometrically with the surrounding scene as well as inpainted RGB content. This is important for accurate optimization of 3D Gaussians for representing the inpainted scene.
3. The unseen mask generation technique identifies occluded regions that require inpainting, leveraging multi-view and depth information to accurately isolate gaps after object removal. This component is crucial for accurate inpainting in complex scenes, where typical object masks might miss crucial occluded areas. It works accurately for many of the indoor and outdoor settings in 360-USID.
4. The method outperforms strategies involving a naive combination of 2D inpainting with 3DGS optimization, producing 3D-consistent inpainted scenes.

**Weaknesses:**

1. Confusing Figure: The method figure (Figure 2) is confusing. For 2(b), why does the inpainting for view k result in a weird cartoonish face? Reference-guided inpainting is supposed to be better than 2(a), which is just per-image inpainting and would suffer from multiview consistency. For 2(c), why does the inpainting mask change from the plant mask in 2(a) and (b) to a small hole in the ground? Even with a simpler inpainting task, views n and k do not quite look multiview consistent. The ground texture is darker in view k.
2. Relevant Baselines missing: InNeRF360 (Wang et al., 2023b) is a very relevant and robust baseline for this method that can perform text-based object removal and subsequent 3D consistent inpainting for in-the-wild scene captures. They have a multiview-consistent segmentation module that can create view-consistent masks of objects of interest (from text inputs), alleviating the need for the capture protocol of the 360-USID dataset introduced in this paper where the exact locations in 3D are captured twice with and without the object to be removed. The paper is cited several times in the text but not used as a baseline.
3. Comparison with non-generative baselines: Feature 3DGS [1] and Feature-Splatting [2] are two recent techniques for editing 3D Gaussian scenes through text queries leveraging foundational priors like CLIP, SAM, DiNOV2 learned alongside Gaussian attributes during training with posed multiview images. As language fields are embedded throughout the scene, these methods can identify Gaussians belonging to a specific object, remove them, and re-render the scene from novel views. Even though their pipelines have no generative inpainting, they serve as strong alternative baselines to evaluate the effectiveness of diffusion-based inpainting.
4. Issues in Figure 7: The ground truth image is not zoomed in, whereas the reconstructions for 360-InpaintR and related baselines are zoomed in to the masked area. This makes judging individual methods unnecessarily complicated. For example, the columns “Gaussian Grouping” and “Ours” look pretty similar, yet we have no idea which one is closer to the ground truth.
5. 3D-Inconsistent Unseen Mask Generation: The unseen mask generation algorithm is supposed to create 3D consistent object masks; however, some of the videos in the supplementary, especially “cookie_compressed.mp4,” do not support the claims.
6. Engineering over Novelty: 360-InpaintR proposes an efficient overall pipeline for reference-guided 3D inpainting but combines several existing techniques. Left Refill is used out-of-the-box for reference-based inpainting, Depth-Anything-V2 for depth inpainting, and the 3DGS optimization relies on the masked LPIPS loss from SPIN-NeRF. However, the strategy for spawning new Gaussians based on the inpainted depth map seems novel, which ensures that 3D Gaussians are accurately added to the scene, respecting existing 3D spatial relationships.
7. [Minor] There are videos for the MipNeRF360 scenes in the supplementary, but NVS results are not included in the paper, even though they are part of the primary evaluation. Only results from the newly introduced 360-USID dataset are included.


References:

[1] Feature 3dgs: Supercharging 3d gaussian splatting to enable distilled feature fields, Zhou, Shijie and Chang, Haoran and Jiang, Sicheng and Fan, Zhiwen and Zhu, Zehao and Xu, Dejia and Chari, Pradyumna and You, Suya and Wang, Zhangyang and Kadambi, Achuta, CVPR 2024.

[2] Feature Splatting: Language-Driven Physics-Based Scene Synthesis and Editing, Ri-Zhao Qiu, Ge Yang, Weijia Zeng, Xiaolong Wang, ECCV 2024

**Questions:**

1. Is there any specific reason behind proposing the 360-USID dataset if there are already techniques for robust text-based 3D-consistent mask generation for object removal/manipulation? The motivation was not clear to me from the writing.

2. Masks for scenes in MipNeRF360 / NeRFStudio: As mentioned in 5.1, scenes of MipNeRF360 / NeRFStudio, ground truth object masks are not available with these datasets, so I am confused as to how the method runs the unseen mask generation pipeline when there is no scope for text-based object-mask creation? The method overview at the beginning of Section 3 clearly states that RGB images and object masks for n views are required as input. There are inpainted videos for the bonsai and kitchen scenes of the MipNeRF360 dataset in the supplementary, but I am unsure how they were generated.

---

### Official Review · Reviewer_u3Lp · 2024-11-03

**Soundness:** 3
**Presentation:** 3
**Contribution:** 3
**Rating:** 6
**Confidence:** 2

**Summary:**

The paper introduced a 3D inpainting framework named 360-InpaintR,designed to inpaint objects in 360-degree unbounded scenes using a 2D inpainted reference for more consistent multi-view inpainting.
With reconstructed 3DGS scene and multiview masks for inpainting, 360InpaintR first removes masked 3DGS object, and then extracts unseen maps with missing Gaussians for later multiview inpainting. By estimating mono-depth of reference image and aligning this estimated depth with remained background depth, 3DGS can be filled by initializing estimated 3D points.
To enhance multiview consistency, one view is selected to be inpainted as the reference view, which guides multiview inpainting via reference-guided image inpainting technique such as LeftFill. With all inpainted views aligned with the reference and their unseen maps, 3DGS scene is further fine-tuned to fill missing areas while preserving seen regions.
Since it is a specific reference-guided 360-degree 3D inpainting task, authors collected a new evaluation dataset with more diverse scenes for thorough evaluation.

**Strengths:**

1. The work proposed a reference-guided unbounded scene inpainting pipeline that leverages priors from reference-guided inpainting and depth models. The truly unseen mask-based inpainting can reduces the inpaintig inconsistency compared with object masked based approaches.
2. The paper provides an in-depth discussion of various 3D inpainting strategies and the challenges faced in the field. Extensive experiments have show proposed 360-InpaintR outperforms baselines, and ablation studies validate the efficacy of proposed techniques.
3. The evaluation with newly collected dataset 360-USID can contribute to the field of 360 inpainting especially within reference-guided 3D inpainting research community.

**Weaknesses:**

The reference-guided pipeline primarily relies on the capabilities of off-the-shelf 2D depth estimation and inpainting models. Further, the method may lead to unstable performance in scenes with uneven floors when using a single reference.

Please also see concerns in Questions.

**Questions:**

1. In Figure 2, what are the representative previous works for the cases shown in Figures 2(a) and 2(b)?

2. The naive 3DGS is trained using L1 and SSIM loss, whereas Eq.6 utilizes L1 and LPIPS instead. Did the initial 3DGS training (prior to depth-aware initialization) also use the losses defined in Eq.6, or were the original losses employed? Additionally, what is the reason for replacing SSIM with LPIPS during fine-tuning?

3. What inpainting model is used for the reference image?

4. In addition to depth-aware Gaussian initialization, I wonder if a depth loss additional to Eq.6 could improve accuracy of 3DGS fine-tuning.

---

### Note · Authors · 2024-11-13

I have read and agree with the venue's withdrawal policy on behalf of myself and my co-authors.